


# Electron precipitation characteristics during isolated, compound and multi-night substorm events

Noora Partamies[1,2], Fasil Tesema[1,2], Emma Bland[1], Erkka Heino[1], Hilde Nesse Tyssøy[2], and Erlend Kallelid[3,1]

[1]The University Centre in Svalbard (UNIS), Norway
[2]Birkeland Centre for Space Science, University of Bergen, Norway
[3]Norwegian University of Science and Technology (NTNU), Norway

**Correspondence:** Noora Partamies (noora.partamies@unis.no)

**Abstract.** A set of 24 isolated, 46 compound and 36 multi-night substorm events from the years 2008–2013 have been analysed in this study. Isolated substorm events are defined as single expansion–recovery phase pairs, compound substorms consist of multiple phase pairs, and multi-night substorm events refer to recurring substorm activity on consecutive nights. Approximately 200 nights of substorm activity observed over the Fennoscandian Lapland have been analysed for their magnetic disturbance

magnitude and the level of cosmic radio noise absorption. Substorm events were automatically detected from the local electrojet index data and visually categorised.

    We show that isolated substorms have limited lifetimes and spatial extents, as compared to the other substorm types. The average intensity (both in absorption and ground-magnetic deflection) of compound and multi-night substorm events is similar. For multi-night substorm events, the first night is rarely associated with the strongest absorption. Instead, the high-energy

electron population needed to cause the strongest absorption builds up over 1–2 additional nights of substorm activity. The non-linear relationship between the absorption and the magnetic deflection at high and low activity conditions is also discussed. We further collect in-situ particle spectra for expansion and recovery phases to construct median precipitation fluxes at energies from 30 eV up to about 800 keV. In the expansion phases the bulk of the spectra shows a local maximum flux in the range of a few keV to 10 keV, while in the recovery phases higher fluxes are seen in the range of tens of keV to hundreds of keV. These

findings are discussed in the light of earlier observations of substorm precipitation and their atmospheric effects.

## 1 Introduction

Substorms are key energy transfer and reconfiguration elements in the magnetosphere–ionosphere system. They use the energy delivered by the solar wind to power a variety of processes in the magnetosphere, which deposit some of the energy into the ionosphere. At the substorm onset, a particle injection from the plasma sheet provides a source population to the ring current

and outer radiation belt region with energies up to tens of keV (e.g. Ripoll et al., 2020). Some of the injected particles end up directly in the ionospheres. This energetic particle precipitation is observed as an intense burst at the substorm onset. These spike events (e.g. Spanswick et al., 2007) are observed as a sharp rise and slow decay of cosmic radio noise absorption (CNA), which moves together with the expanding particle precipitation region in the ionosphere. In the inner magnetosphere, the



injected electrons may undergo further acceleration by wave-particle interaction up to energies of hundreds of keV, and even
MeV. These relativistic particles can be lost in the atmosphere through pitch angle scattering, resulting in a more spread out
high-energy drizzle during the recovery phase after the initial substorm onset. In the ionosphere, this is related to diffuse and
pulsating aurorae, which are primarily observed in the magnetic morning sector (e.g. Nishimura et al., 2020).

Substorms occur both in isolation and in recurrence. These two types were introduced by Borovsky and Yakymenko (2017),
who used the terms "randomly occurring" and "periodically occurring". From a large statistics of substorms detected both
in ground-magnetic data and in particle injection data at geostationary orbit, they concluded that the randomly occurring
substorms have a recurrence time (waiting time) of 6–15 hours, while the periodic ones occur every 2–4 hours. Both substorm
types were found to be associated with enhanced solar wind driving, but no evidence of the recurrence rate driver of the
periodic substorms was identified in the solar wind. Many earlier substorm studies have landed on a similar categorisation of
substorms (e.g. Newell and Gjerloev, 2011; Rodger et al., 2016; Liou et al., 2013; Sandhu et al., 2019), all with substorms
grouped into isolated and recurrent, periodic, non-isolated or compound, each of which with a slightly different meaning, but
always something more active and morphologically more complex than an isolated substorm. The isolated substorms have
systematically been defined to include a few hours of quiet time prior to the onset and largely mean the same from one study
to another. For instance, Liou et al. (2013) concluded that the isolated substorms are intrinsically no different from the non-
isolated, just related to weaker solar wind driving and lower precipitation power. This logic is true for all of the more complex
categories mentioned above.

Substorms can be identified using data from space or from the ground. Automatic substorm detection algorithms often use
the auroral electrojet index, which is a continuous and easily available data set. Whether the index implemented in the detec-
tion is the original global AL index (Juusola et al., 2011), the SuperMAG SML index (Newell and Gjerloev, 2011), or a local
electrojet index (e.g. Partamies et al., 2013) is determined by the purpose of the study. Earlier attempts have incorporated
fixed threshold values for substorm detection criteria, while an important improvement to that is percentile thresholding on the
rate of change (Forsyth et al., 2015). Different detection algorithms share some common basic concepts: substorm onsets are
identified as abrupt decreases in the ground-magnetic north-south (or horizontal) component, expansion phases start from the
onset and last until the minimum of the magnetic deflection, and recovery phases last until the magnetic disturbance has de-
cayed. The actual threshold values for the rate of change of the magnetic field at the onset, the threshold value for the measured
magnetic field minimum, the definition for the end of the recovery, as well as the definition of the growth phase vary between
the different algorithms. The search routines still produce similar statistical results on substorm occurrence and duration, as
concluded by Forsyth et al. (2015), although the individual onset and phase timings can differ. As also pointed out by Borovsky
and Yakymenko (2017), different substorm descriptors hardly give exactly the same set of events. In their study, about 60% of
substorms identified by particle injections coincided with substorms detected by ground-magnetic signatures. Thus, it is very
important to clearly define and choose descriptors that are best suited for the purpose.

Energetic particle precipitation can be monitored from the ground by measuring the absorption of the cosmic radio noise
(CNA) in the atmosphere. CNA has been shown to closely follow the variations in the geomagnetic activity, as demonstrated





by a linear correlation between the logarithm of CNA and Kp index (Kavanagh et al. (2004) and references therein). The

CNA distribution was found to have a local time dependence: The strongest CNA values were seen in the pre-midnight and midnight sectors due to substorm activity, while another high absorption sector was shown to be the morning due to the eastward drift of substorm injected electrons. All CNA and geomagnetic data were included in their study without selecting any specific events, such as substorms, which are associated with strong absorption. The correlations reported by Kavanagh et al. (2004) had coefficients around 0.4, which were speculated to increase if time periods with higher activity were examined

separately. A quadratic relationship was suggested and tested to better describe the relationship between CNA and Kp with a higher correlation of about 0.5. However, in some time sectors (for instance at 15–18 MLT), the dependence between the two parameters was approximately linear. The non-linear contributions were generally assigned to the high activity or very low CNA. An AE index was suggested to be better correlated with CNA as it would capture the rapid substorm-related changes in the magnetic activity more accurately than the 3-hour averaged Kp.

An overview of the particle precipitation during the substorm cycle has been presented by Wing et al. (2013). They divided the precipitation into three different categories: monoenergetic, diffuse, and wave electron aurora. Monoenergetic precipitation refers to field-aligned acceleration, diffuse precipitation relates to pitch angle scattering of plasma sheet electrons in resonance with very low–frequency whistler mode chorus waves, and wave electron aurora is associated with electron interaction with dispersive Alfvén waves. All the different precipitation types strongly increase at the substorm onset: monoenergetic by 70%,

diffuse by 300%, and wave electron aurora by 170%. The substorm onset was detected simultaneously at all precipitation types, but the diffuse precipitation took 1–2 hours longer to decay than the other types of electron precipitation. The mechanism generating the diffuse precipitation is the one that most strongly contributes to the high-energy electron precipitation and, thus, to the evolution of CNA. Although this statistical study was based on a decade of particle data from a spectrometer onboard the Defence Meteorological Satellite Program (DMSP) spacecraft, where most of the energy channels count electrons with

energies below 10 keV, the tripling of the precipitation power indicates a significant enhancement in the high-energy electron precipitation as well. Furthermore, it is important to note that the diffuse precipitation is a particularly large-scale feature: after the onset, it statistically occupies 10–12 hours of magnetic local time at the magnetic latitudes of about 65–70 degrees. Furthermore, the lifetime of the diffuse precipitation exceeded the epoch time used in their analysis (about 2 hours after the onset).


A detailed study of substorm electron precipitation spectra by Beharrell et al. (2015) presented a model for injections of energetic electrons at the energies of 20–300 keV (so-called medium-energy electrons). They used the substorm list by Newell and Gjerloev (2011) as their onset times, as well as typical substorm parameters found in the literature. These included the substorm onset location close to the magnetic midnight, the flux of the injected particles, the energy spectrum of the injection,

the temporal evolution of electron flux, and the mean lifetime of injected electrons (for details and references, see the model description in their paper). The number of energetic electrons injected during substorms was used to estimate electron density profiles for the ionospheric region affected by the substorm injection, which was further used to calculate the absorption due to the particle precipitation. The model absorption was then matched to the measured CNA to obtain the best possible





flux magnitudes of the particle injections for a 5-day period of 61 substorm onsets during a mild geomagnetic storm (Dst
index between -40 and -10 nT). The energetic electron precipitation (EEP) forcing described by this sequence of modelled
substorms was further used to estimate the corresponding atmospheric impact (Seppälä et al., 2015). The one-dimensional
Sondakylä Ion and Neutral Chemistry model (e.g. Turunen et al., 2009) was run to investigate the production of odd hydrogen
($HO_x=OH + HO_2$) and odd nitrogen ($NO_x=N + NO + NO_2$) due to the strong ionisation at the bottom part of the ionosphere,
and the following catalytic depletion of mesospheric ozone. The peak loss of mesospheric ozone was found to be 10–50%
depending on the season. Due to the lower background ionisation level, the winter solstice substorm forcing penetrated deeper
in the atmosphere, causing stronger and longer–lasting ozone loss. The maximum depletion was observed during day 3 and 4
in the substorm sequence, when the recurrence rate also maximised. The range of modelled ozone depletion was estimated to
be comparable to the ozone loss of a small to medium solar proton event (in the range of about 500–6000 pfu, von Clarmann
et al., 2013).

The substorm recurrence rate of about 15 onsets a day (Beharrell et al., 2015) is much higher than any longer-term average
rate of a few per day (e.g. Borovsky and Yakymenko, 2017). At the same time, the substorm detection routines have slightly dif-
ferent definitions and threshold values, and the previous studies have indicated both temporal delays and non-linear magnitude
dependence between the magnetic disturbances and the energetic particle precipitation signatures, as outlined above. Our aim
with this study is to investigate whether all substorms can be equally influential to the atmosphere, or whether some events can
be ignored based on their intensity, duration or internal structure. In this paper, we use the cosmic noise absorption signature
as a proxy for the medium-energy electron precipitation. We investigate isolated, compound, and multi-night substorm events
with different magnetic disturbance magnitudes, examine any cumulative effects the multi-night events may have on CNA, and
compare in-situ particle precipitation spectra to previously observed reference values of energetic electron forcing.

## 2 Data

### 2.1 Magnetic activity indices and substorm phase detection

The lower envelope curve of the global auroral electrojet index (AL) is sensitive to enhancements in the westward electrojet,
which makes the AL index a good tool for identifying substorm signatures. An AL index based substorm phase detection
method was introduced by Juusola et al. (2011). The method detects start and end times of substorm phases using the following
criteria:

1. The growth phase begins from the IMF $B_Z$ southward turning and ends at the substorm onset

2. The substorm onset is an abrupt decrease in the AL index with the rate of change of at least 4 nT/min.

3. The expansion phase begins at the substorm onset and ends at the AL index minimum, which must be less than -50 nT.

4. The recovery phase begins at the AL index minimum and lasts until the AL index has reached values above -50 nT, or
   until a new onset.





The threshold value of -50 nT comes from a long-term median of negative AL index values. OMNIWeb solar wind data, which has been propagated to the magnetopause, is used to determine the IMF $B_Z$ polarity.

Juusola et al. (2011) validated these substorm detection criteria against a list of substorm onsets published by Frey et al. (2004) and concluded that the agreement was good. Instead of the global electrojet index (AL), Partamies et al. (2015) used a regional electrojet index (IL version they called $IL_{ASC}$) constructed from baselined data collected at five Lapland magnetometer

stations of the IMAGE network (Tanskanen, 2009): Kevo (KEV, 69.76°N, 27.01°E), Kilpisjärvi (KIL, 69.02°N, 20.87°E), Muonio (MUO, 68.02°N, 23.53°E), Abisko (ABK, 68.36°N, 18.82°E), and Sodankylä (SOD, 67.42°N, 26.39°E). The selected stations were co-located with the MIRACLE auroral all-sky cameras (Sangalli et al., 2011), allowing the auroral morphology to be analysed over the same area. They also concluded that the long-term median value of -50 nT was valid for this regional index as well. Furthermore, it is important to note that an earlier study by Kauristie et al. (1996) shows that a local electrojet

index, IL (including the entire IMAGE magnetometer network), corresponds well to the global AL index in the magnetic midnight sector (20–02 UT in Lapland). A more recent study by Tanskanen (2009) suggested that the reliable time range could be extended to 16–03 UT. Since the magnetic midnight sector is the most favourable time range for substorm activity and the Lapland latitudes are most of the time under the substorm activity (e.g. Frey et al., 2004), results from nighttime substorm studies over Fennoscandian Lapland should be globally applicable. In this study, we use the Lapland substorm phases detected

by Partamies et al. (2015) as a starting point. Thus, the IL index used in this study refers to the regional Lapland index $IL_{ASC}$ throughout the paper. A further visual selection and sub-categorisation of events will be described in section 2.4.

## 2.2 From cosmic noise absorption to a regional absorption index

Measurements of cosmic radio noise absorption (CNA) from a chain of riometers owned and operated by the Sodankylä Geophysical Observatory (SGO) have been used here to describe the substorm EEP impact in the D region ionosphere. Increased

ionisation in the D region leads to enhanced absorption of the cosmic noise, and at D region heights this is mainly due to precipitation of electrons with energies above 10 keV (e.g. Turunen et al., 2009). The SGO riometers are wide-beam instruments which listen to the cosmic noise at approximately 30 MHz. CNA is calculated as the reduction of cosmic noise with respect to the quiet background, the so-called Quiet Day Curve (QDC). For the SGO riometer data, the QDC is calculated automatically by fitting a sinusoidal curve to the data of the ten previous days. Our automatically baselined (or QDC subtracted)

dataset extends from 2008 until 2013 with a 1-minute temporal resolution. For more instrument details, see for instance Heino & Partamies (2020). For this study, we selected the Lapland riometer stations at Ivalo (IVA, 68.56°N, 27.29°E), Abisko and Sodankylä to match the magnetometer stations used for the local electrojet index. We further calculate an "absorption index" by aligning the baselined CNA data from the three stations and taking the upper envelope curve, similar to the construction of the global AU or the local IU index. Together with the regional electrojet index (IL), this absorption index allows us to capture

the magnetic disturbances and the particle precipitation enhancements occurring within approximately the same geographic area over the same time period. Note, that the terms "absorption" and "CNA" in this paper refer to this *regional CNA index*.



## 2.3 Space-borne particle precipitation measurements

To characterise the particle precipitation energy spectra during the substorm events, we searched for overpasses of the low-altitude spacecraft, Defence Meteorological Satellite Program (DMSP) and Polar Orbiting Environmental Satellites (POES). Together the spacecraft from the two satellite programs cover electron energies from 30 eV up to almost 800 keV. The upward-looking spectrometers (SSJ versions 4 and 5) onboard DMSP measure fluxes of downward-going electrons at 19 energy channels from 30 eV up to 30 keV (Redmon et al., 2017).

POES observes particles with two different instruments, the Total Electron Detector (TED) and the Medium Energy Proton and Electron Detector (MEPED). TED measures differential electron fluxes in four energy bands (0.15–0.22, 0.69–1, 2.12–3.08 and 6.50–9.46 keV) with telescopes pointing up and at 30° to the vertical (Evans & Greer, 2000). We used data from the upward–pointing telescope only, which may lead to an underestimation of the precipitation electron fluxes.

Similarly, the MEPED instrument comprises two telescopes, one pointing upwards and another one normal to the first. The measurements consist of fluxes of four integral channels (above 43, 114, 292 and 756 keV, Ødegaard et al., 2017). For these data, we used a combination of the measurements from both telescopes to construct the bounce loss cone fluxes as described by Nesse Tyssøy et al. (2016). Their pre-processed dataset further includes corrections for proton contamination in the electron channels and identification of the relativistic electrons on the proton detector.

To make the MEPED integral fluxes comparable to the TED and SSJ differential fluxes, we converted the observed integral fluxes into differential fluxes. The resulting three flux values are set to the centre energy between the integral channel cutoff energies of 78.5, 203, and 524 keV. The data and the approach are similar to that described by Tesema et al. (2020), except for the extrapolation of the MEPED spectra into lower and higher energies, which we consider unnecessary for the purpose of the current study. All particle data analysed in this study are in the format of overpass-averaged spectra, also used by Tesema et al. (2020).

## 2.4 Event selection and substorm categories

The event categorisation was performed visually using nightly overview plots similar to Figures 1 and 2. The middle panel of each figure shows the temporal evolution of the CNA index, and the bottom panel shows the IL index. The green, red and blue shadings mark the time periods of automatically detected growth, expansion, and recovery phases respectively. As the automatic substorm phase detection routine is also sensitive to small events (thresholded by the IL index long-term median value of -50 nT), a visual inspection was done to exclude events with IL minimum above -300 nT. These mild events are generally not accompanied by an appreciable CNA enhancement. We required two days of quiet time (no automatically detected substorms) prior to all of the event groups described below. This is done to make sure that the activity starts from a solidly quiet background, which allows us to determine whether there is a cumulative ionospheric response to the energetic particle precipitation.

*Isolated* substorms are defined as events with a single expansion–recovery phase pair. An example of an isolated substorm is plotted in Figure 1: the CNA (middle) and IL (bottom) index evolution with colour shadings for substorm phases. The keogram



from the SOD camera station in the top panel shows that the auroral green emission (557.7 nm) evolves in tandem with the magnetic disturbances and the absorption enhancements. The expansion phase contains the brightest emission, while diffuse emission is seen in the recovery phase.

*Compound* substorms are defined as consecutive expansion–recovery phase pairs that are not interrupted by quiet time or a growth phase. A similar definition was employed by Sandhu et al. (2019). In case of compound substorms, the substorm

onset is the beginning of the first expansion phase. Later expansion phases are called intensifications. Although the threshold for the IL index minimum is -300 nT, we allow an intensification in the middle of the substorm activity to be as small as -100 nT, as long as the lifetime of the expansion exceeds 20 minutes. All these threshold values are, of course, somewhat arbitrary, but they are based on our visual comparisons of the magnetic and absorption signatures from hundreds of events. An example of a compound substorm is shown in Figure 2. This event contains two expansion–recovery phase pairs, as well as

a non-detected intensification (IL dip of about 200 nT at 22:04 UT). Prior to the visually approved onset (first black vertical line), a minor substorm event took place. This event was excluded from further analysis due to its low intensity (IL > -300 nT) and a short growth phase between this minor event and the major onset at 19:50 UT. Another minor event that occurred the following morning, well after the main activity, was also ignored. The top panel in Figure 2 again shows the auroral evolution from the SOD camera station as a keogram. The largest expansion (the second red shading in between the black vertical

lines) corresponds to both enhancement in the CNA and bright aurora, while the first auroral brightening happened during the excluded event prior to the main event onset at 19:50 UT. The third auroral brightening occurred at around 23:00 UT coinciding with the second major CNA enhancement but deep into the magnetic recovery without any appreciable IL index intensification.

*Multi-night* substorm events are defined as substorm activity that occurs on consecutive nights. The individual nights during these events consist of either substorms with a single–phase pair or substorms with multiple intensifications. They can look

like isolated or compound substorms (as described above) during any of the individual nights, but after a magnetically calm daytime (which is excluded from the analysis), the activity resumes for one or more additional nights. Each night has its own substorm onset, and may include one or more instensifications. In total, the 36 multi-night events include 134 individual nights of substorm activity, most of which were linked into a series of 3–4 nights of activity, but a handful of events was found to continue over 6–7 consecutive nights.

A summary of the identified and categorised events is given in Table 1. Each event duration excludes the growth phase prior to the substorm onset. Each event has been assigned an intensity value, which is the minimum IL index value rounded to the closest 100 nT. The median value of the substorm intensity in all three groups is around -500 nT (not included in the table). However, the range is very limited for isolated (from -300 to -800 nT) and compound events (from -300 to -900 nT), while it becomes much larger for the multi-night events (from -300 to -1800 nT). It is important to note that in the group of isolated

substorms, there is only one event reaching the extreme IL value of about -800 nT. Based on our set of events, a substorm negative bay with IL minimum below about -600 nT is highly uncommon for isolated substorms starting from quiet conditions. The durations and Dst indices given in Table 1 are median values for each substorm type. The range of Dst index values is larger in the group of multi-night events than it is in the other two sub-categories, but as the median values suggest, the typical events are related to very similar ring current enhancements in the groups of compound and multi-night substorm events. The number



**Figure 1.** An example of an isolated substorm event with a single expansion (red) and recovery (blue). The vertical lines mark the time period which is used in the analysis, i.e. from the expansion onset until the end of the recovery phase. The green shaded area marks the growth phase as determined by the search algorithm. Displayed is the temporal evolution of CNA (middle) and IL index (bottom). The top panel shows the auroral evolution at 557.7 nm as a keogram (north–south slice as a function of time) from the SOD camera station aligned to match the timing of data in the two bottom panels.





**Figure 2.** An example of a compound substorm event with a several consecutive expansion (red) and recovery (blue) phases. The vertical lines mark the time period which is used in the analysis, i.e. from the expansion onset until the end of the recovery phase. The green shaded area marks the growth phase as determined by the search algorithm. Displayed is the temporal evolution of CNA (middle) and IL (bottom). The top panel shows the evolution of the green auroral emission as a keogram (north–south slice as a function of time) from the SOD camera station aligned to match the timing of data in the two bottom panels.





**Table 1.** A selection of descriptive parameters to characterise the different substorm categories.

| Event type | # events | duration (h) | #phase pairs | Dst (nT) |
|---|---|---|---|---|
| Isolated | 24 | 2 | 1 | -9 |
| Compound | 46 | 5 | 2–3 | -24 |
| Multi-night | 36 | 5 | 3–4 | -25 |

of phase pairs is the number of automatically detected expansion–recovery phase pairs during the event, where phases shorter than 20 minutes have been ignored (as described above). In the group of multi-night events, the table shows the number of phase pairs per night. The number of phase pairs varies between 2 and 9 from one event to another in the groups of compound and multi-night substorms. The highest numbers of phase pairs are found during multi-night substorm events.

## 3 Results

### 3.1 The relationship between CNA and magnetic disturbances

Since the correlation between magnetic disturbances and absorption has been studied before, we want to determine the extent to which our dataset follows the previously established relationship. Figure 3 is a scatter plot showing the minimum IL index and the absorption for each event. The events are colour-coded as red, blue and black for isolated, compound and multi-night events respectively. Values for multi-night events describe individual nights. Although the Pearson correlation coefficient for 235 the full dataset here is -0.6 (with $p < 0.01$), a number of obvious outliers can be seen in the figure. The correlation in the isolated substorm category is insignificant, while significant correlations ($p < 0.01$) are found in the categories of compound and multi-night events with correlation coefficients of -0.5 and -0.7, respectively.

The temporal evolution of CNA during the substorm events from 3 hours before to 7 hours after the onset is illustrated by the superposed epoch plots of CNA in Figure 4. The typical evolution of the isolated substorms shows a mild maximum ($\sim$0.5 dB) 240 in the median curve (blue) at the substorm onset time, which then decays during the following two hours. The two other groups of more complex substorms show a slightly higher CNA enhancement in the median curve (up to $\sim$0.6 dB) at the onset time, which does not recover within the seven-hour time frame shown here, not even in the 25% percentile curve (bottom red curve). Interestingly, the top percentile for both of these event groups maximises an hour after the onset, as well as 4–6 hours after the onset, which is most likely a signature of multiple substorm intensifications. Note that there is no appreciable CNA difference 245 between the compound and multi-night events. This is probably due to the multi-night events being a mixture of nights with short single phase pair substorms and those with multiple phase pairs and long durations.

Figure 5 shows the evolution of the median CNA for eight different IL index intensities from IL$= -300$ nT to IL$< -900$ nT. In general, CNA increases with increasing susbtorm intensity. For the weakest electrojet intensities (IL$\geq -500$ nT), the maximum CNA occurs at the zero epoch. In contrast, for events with IL$\leq -900$ nT the peak CNA is delayed by one hour (top

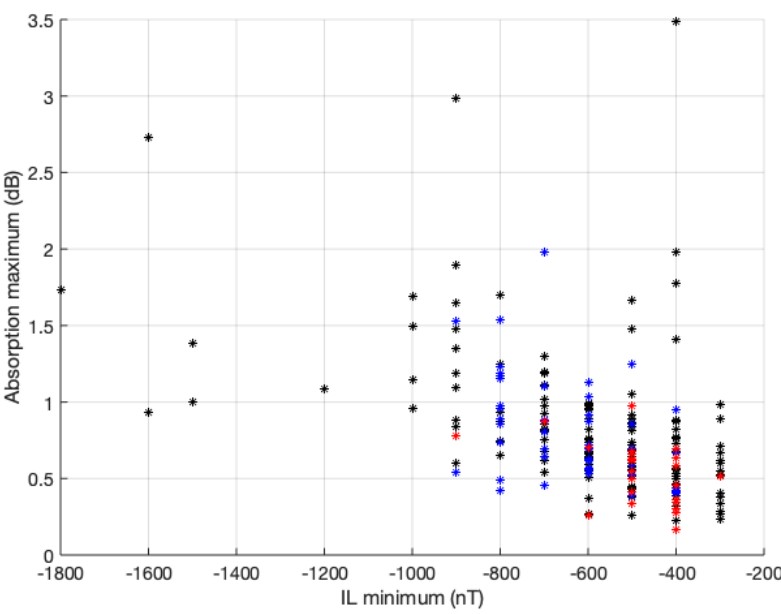

**Figure 3.** A scatter showing the correlation between the CNA and IL index values for the substorm events. The events are separated into the groups of isolated (red), compound (blue) and multi-night (black) events.

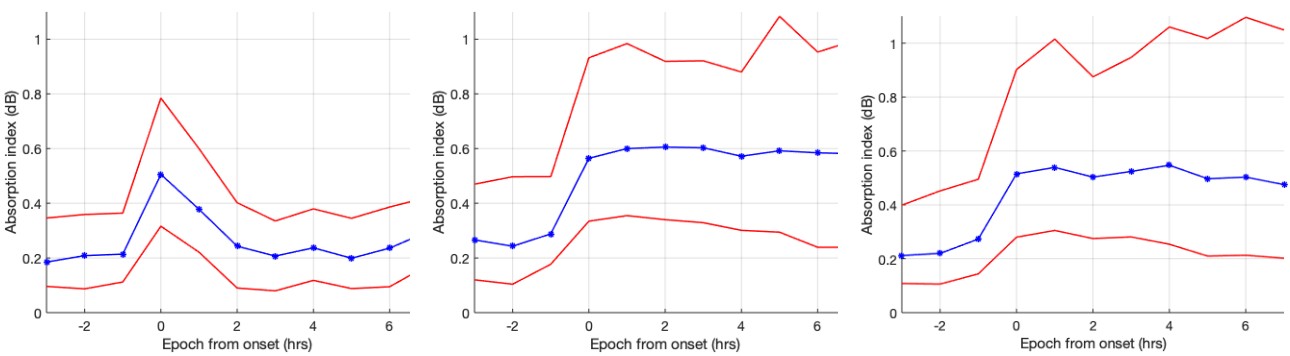

**Figure 4.** A superposed epoch view to the isolated (left), compound (middle) and multi-night (right) substorm events from -3 to +7 hours from the onset of the first expansion phase. For multi-night events, the onset is the start of the first expansion phase of each night. The median curves are blue and the 25% and 75% percentile curves are red.

250 two curves). For these stronger events the temporal evolution of the CNA is also highly variable, and they are less obviously ordered by the IL intensity.

We further investigate the CNA evolution during the multi-night substorm events, in particular, how the CNA responds to the IL change during the different nights in a series substorm activity. These results (not shown) suggest that CNA often grows





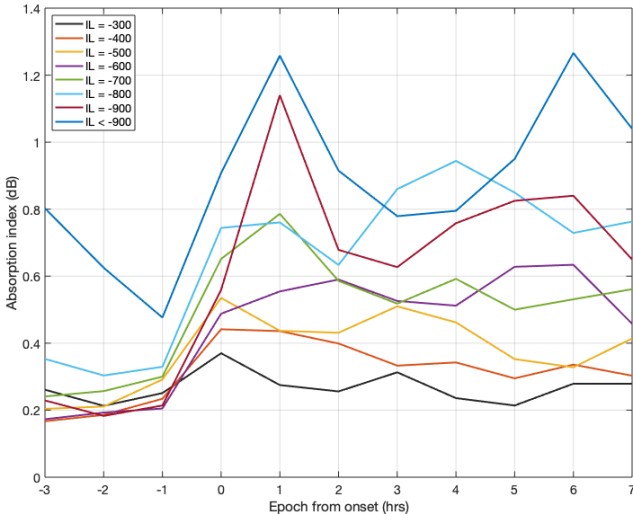

**Figure 5.** Median CNA curves for different ranges of IL index values without any substorm type categorisation. The intensity categorisation results in 10–40 events per group. The data are binned into 1-hour resolution.

strongest during the second, third or fourth night, and is less dependent on the IL intensity of that night, which may account
for some of the temporal CNA variability in Figure 5. The most intense absorption is rarely observed during the first night of
activity. To illustrate the mismatch between the IL index and CNA index behaviour during the multi-night activity Figure 6
shows the evolution of the IL (left) and CNA (right) intensity for the multi-night events. The intensities for IL and CNA are
plotted with respect to the intensity values measured during the first night; thus each event curve starts from 0. In total, there
were 23 cases where the activity continued for three or more consecutive nights, 14 cases with 4 or more consecutive nights,
and 7 cases with 5 or more consecutive nights. While the IL index evolution (left panel) shows a well-balanced distribution of
positive and negative values during the second to fourth night (i.e., increase and decrease with respect to the first night) with
average values within ±40 nT, the CNA evolution (right panel) is biased towards positive values during the second to third
night with average values of 0.6–0.7 dB. In about 70% of the cases, CNA increases from night 1 to night 2, with a median
enhancement of 0.7 dB.

**3.2   Particle precipitation spectra**

During the 204 detected and categorised substorm periods, we found 30 DMSP overpasses and 124 POES overpasses in total.
Most of the overpasses (123) took place during the substorm recovery phases, which is expected because those are the longest-
lasting substorm phases, while 31 overpasses coincided with expansion phases. The distribution of the overpasses between
the different substorm types is very uneven, with only 3 overpasses found during the isolated, 34 during compound and 117
during the multi-night events, which reflects the large differences in the lifetimes of the event types. To account for this uneven

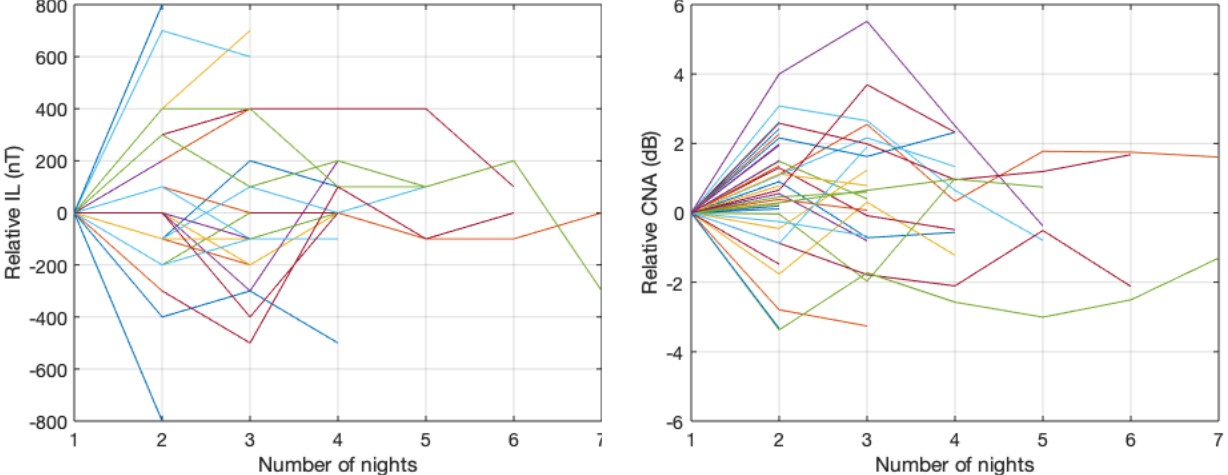

**Figure 6.** An event-by-event evolution of the IL index (left) and the CNA index (right) during multi-night events. Each event is represented by one curve, whether the event lasts for two or seven nights in a row. The minimum IL value and the maximum CNA value reached during the first night has been subtracted. As the IL minima have been rounded to the nearest full 100 nT some of the event evolution lines appear on the top of each other, making the plot look less busy than the CNA plot, although the number of events is equal in both panels.

distribution of overpasses, we group all expansion phase data into one bundle of spectra, and all recovery phase data into another, as shown in Figure 7. During the expansion phases (left panel), a majority of the DMSP overpasses (blue) shows enhanced electron fluxes at energies of a few keV. These fluxes are higher than the boundary fluxes of the pulsating aurora (black dashed curves according to Tesema et al., 2020). Pulsating aurora (PsA) is a type of diffuse aurora which occurs mainly

in the late recovery phases of substorms (Partamies et al., 2017). Expansion phase fluxes are found at and around the PsA upper boundary up to about 10 keV, while fluxes at higher energies depart from the PsA upper boundary, and the median flux curve (solid black line) resides well inside the PsA band. During the recovery phase overpasses (right panel), the flux values are much more variable at all energies. The fluxes up to 10 keV are mostly confined between the PsA boundaries, while the fluxes at and around 100 keV cluster in the upper part of the PsA band, close to the PsA upper envelope curve. Compared to

the expansion phase fluxes, the recovery phase fluxes at energies below 10 keV are lower by half an order of magnitude, as indicated by the median flux curves (solid black). On the other hand, the fluxes at higher energies are slightly enhanced during the recovery phases as compared to the expansion phases.

Similarly, we compare the precipitation spectra for events with one or two phase pairs to those with five or more phase pairs (spectra not shown). The fluxes at and around 100 keV are about an order of magnitude higher for events with five or more

expansion phases than for those with only 1–2 phase pairs. Furthermore, during the first night of the multi-night events, the median flux at and around 100 keV is lower than during the second night, by about half an order of magnitude. These findings are in good agreement with the CNA results described in the previous section.

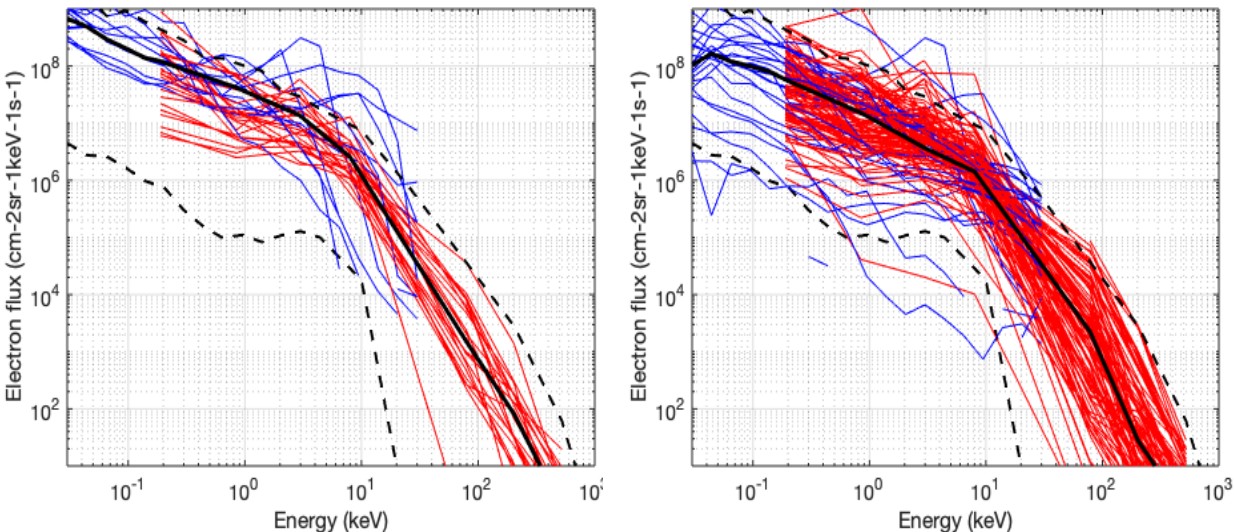

**Figure 7.** Precipitation spectra for expansion phases (left) and recovery phases (right) from overpasses of DMSP (blue) and POES (red) spacecraft. The dashed curves are the upper and lower boundary spectra for pulsating aurora from Tesema et al. (2020), and the solid black curves are the median fluxes of our substorm events.

## 4  Discussion

Isolated, compound, and multi-night substorm events have been analysed with respect to the magnitude of their magnetic
disturbance (IL index minimum) and the related cosmic noise absorption (CNA maximum). About 100 substorm events over the course of 6 years were automatically detected and visually classified. The substorm detection algorithm used in this study has provided well-grounded results in earlier large statistical analyses (Juusola et al., 2011; Partamies et al., 2013). Since the IL index threshold value in this method allows the detection of very mild events which do not produce an appreciable CNA enhancement, we have visually pruned the events to exclude all cases with magnetic deflections less than -300 nT. We have
also required two days with no detected substorms prior to any of the events included in this study. This makes it less likely that there would be a significant high-energy particle storage in the radiation belt region, which could be tapped by a solar wind pressure pulse or another solar wind transient, as the nominal loss time for radiation belt particles with energies of the order of 100 keV is from hours to about a day (Summers et al., 2008). Thus, after two days of no substorm injections the particle storage build-up starts from the quiet magnetospheric conditions. This criterion is different from earlier substorm studies, which mostly
require a quiet time of about 3 hours prior to isolated substorms. As pointed out by Sandhu et al. (2019), a stronger solar wind driving was often maintained for several days prior to the compound substorm onsets.

It is important to note that most of the studied substorm events are not storm-time substorms, as indicated by the mild median Dst values of about -20 nT. For the multi-night events, which would be the candidates for the strongest magnetic





activity, the median solar wind speed was around 530 km/s. The top 25% of the events were driven by the wind speed at and above 600 km/s. Kavanagh et al. (2012) defined a HSS as a period of sustained solar wind speed above 500 km/s, and concluded that these conditions typically cause a period of multi-day substorm activity and longer-lasting enhancements in CNA. The sequence of 61 substorms (Beharrell et al., 2015), which was shown to lead to a significant mesospheric ozone depletion (Seppälä et al., 2015), took place within a Dst variation between -40 and -10 nT. The activity was driven by a HSS

with the solar wind speed over 600 km/s for about 2.5 days (the driver conditions were not analysed in the paper). This period of high solar wind speed also coincided with the period of the highest substorm rate and the largest ozone depletion. About half of the individual nights during our multi-night events fall into this category of weak magnetic storms driven by the fast solar wind. The bias towards mild geomagnetic activity in our study is likely to be caused by the quiet time requirement prior to all types of events.

As illustrated by the example events in Figures 1 and 2, the isolated substorm events are often not only short-lived compared to the other two types, but also limited in their spatial coverage. The localisation of the isolated event can be seen in the ASC keogram (top panels) where the auroral emission is limited to about half of the all-sky view (Figure 1), while during the compound substorm event the auroral emission moves from the northern to the southern horizon over the course of the activity. For consistency, we visually investigated the ASC data for all substorm events. About 40% of the individual nights

of substorm activity took place in summertime outside the auroral imaging season. Cloudiness prohibited auroral observations for approximately half of the remaining events. Thus, for about 30% of the events, ASC data could be examined. This set of events included 10 isolated, 10 compound and 42 individual nights during multi-night substorm events. Similarly to the sample substorm presented in Figure 1, the auroral emission during the rest of the isolated substorms were also found not to occupy the full ASC field-of-view at any of the Lapland stations. This suggests that in order to reliably use the spacecraft

particle precipitation data to construct an average precipitation spectrum for isolated substorm events, the spatial extent should be assessed carefully to exclude overpass time outside the substorm particle precipitation region. The optical emission during the compound substorms was often found to be discontinuous, even though the magnetic disturbances were not interrupted with quiet time or substorm growth phases. The multi-night events were most often associated with a large spatial extent of auroral emission, which filled an individual ASC field-of-view. In fact, auroral emission in many of these events extended

from Lapland to the southern part of the Svalbard archipelago, being observed in the ground-based auroral images there. The inner magnetospheric observations by Sandhu et al. (2019) showed that the energy content in the ring current was azimuthally more localised during isolated than compound substorms, leading to smaller-scale substorms. However, they concluded that the latitude (L shell) extent of the enhancements varied less, suggesting that the same would apply to the high-energy part of the substorm precipitation. Gjerloev et al. (2007) reported on a full-width half-maximum extent of 3 hours in MLT and

4.7° in latitude of 116 substorms studied using global auroral images, which is similar in latitude extent but narrower in MLT range than the extent of diffuse aurora (Wing et al., 2013). Thus, it is not clear if the spatial extents of either the optical auroral emissions or the magnetic disturbances during substorms are a good proxy for the impact area of high-energy particle precipitation. A more direct estimate of the atmospheric impact area could be obtained by determining the spatial extent of the CNA, since this parameter is sensitive specifically to the particle precipitation energies associated with an atmospheric impact.





Bland et al. (2020) recently used the Super Dual Auroral Radar Network (SuperDARN, Lester, 2013) to estimate the spatial coverage of the particle precipitation impact area during pulsating aurora by studying the attenuation of 10–12 MHz radio noise in the D region ionosphere. They found that the atmospheric impact area extended over at least 4° of magnetic latitude for 75% of the PsA events studied, and 36% of the events extended over at least 12° of magnetic latitude. This method could be used to constrain the spacecraft energy spectra measurements to the atmospheric impact area during the substorm expansion

and recovery phases, in addition to, or even instead of the optical auroral data, and hence construct a more realistic description of the substorm precipitation spectrum.

The CNA values during the substorms studied here peaked between 0.5 and 1.0 dB (Figure 4), which is between the long-term average CNA including the quiet times (CNA mainly below 0.5 dB, Kavanagh et al., 2004) and the absorption values related to geomagnetic storms driven by coronal mass ejection (CME) sub-structures, sheath and ejecta (CNA values typically

around 1 dB, Kilpua et al., 2020). An earlier study on HSS-driven substorms reported CNA levels of 1–2 dB (Grandin et al., 2017). Thus, CNA values tend to increase with increasing magnetic activity on average. In agreement with the previous studies, we show a linear relationship between the magnetic disturbances and the CNA values (Figure 3). We also find a similar level of correlation between the two parameters, although we only examine substorm events without including the quiet time values. Some obvious departures from the linearity, as also pointed out by Kavanagh et al. (2004), were related to the high/low activ-

ity values. The saturation of the absorption when the IL deflection grows beyond -800 nT leads to poor correlation. Another poorly-correlated substorm group is the isolated, less intense, and short-lived events, which do not typically cause strong CNA. Hence, the short lifetime and weak D region ionisation imply that these events will have a minor impact on the atmospheric chemistry. Compound and multi-night substorms will, however, be significant contributors to the direct production and $HO_X$ and $NO_X$ radicals in the atmosphere. Although some of the nights during multi-night substorm events can have mild IL in-

dices, the corresponding absorption values may still be significant. These findings emphasise the atmospheric influence of the compound and multi-night substorm events, which may explain why the global geomagnetic activity indices serve us so well as energetic particle precipitation proxies, despite their poor temporal and spatial resolution.

In total, 70 events included CNA enhancements during the growth phases prior to the onset. These signatures are similar to

the growth phase CNA observations reported by, e.g., McKay et al. (2018) and Sergeev et al. (2012). The large majority of events with growth phase CNA signatures (66 events) were compound or multi-night events. In the case of multi-night substorm events, only 3 events included a growth phase CNA enhancement during the first night. Every isolated substorm with a CNA enhancement prior to onset was more intense than the average of -500 nT. This suggests that the growth phase absorption typically requires magnetospheric preconditioning in the form of prior substorm activity. These CNA values are not counted

towards the CNA averages presented here, as we have excluded the growth phases from our analysis. However, the previous studies show that growth phase CNA enhancements are not always present, their durations are limited to the minutes and tens of minutes prior to onset, and they are confined to a limited latitude region equatorward of the growth phase arc. They are, therefore, unlikely to sum up to a large global atmospheric impact.





Our results suggest that the strong CNA and energetic electron precipitation needs some "cooking time" to build up the favourable conditions in the inner magnetosphere. This is supported by the finding that the first onset/phase pair (or the first night in multi-night events) is rarely the strongest one. Contrary to the multi-day events discussed by Kavanagh et al. (2012), our multi-night substorm sequences are typically not causing the strongest CNA during the first night of activity but rather during second-to-fourth night. The mildness of the first night in our study probably relates to the 2-day quiet time requirement

prior to the substorm events. The in-situ particle data from POES MEPED show a similar delay in the M–I coupled response independently, in that the flux of high-energy ($\gtrsim$100 keV) particle precipitation is lower during the first night of multi-night events than during the later nights of multi-night events. Thus, the cooking time may be required to accelerate electrons in the inner magnetosphere through wave-particle interactions. This conclusion agrees well with earlier observations of the radiation belt dynamics, where a 1–3 day delay was found between the substorm activity and the maximum radiation belt response

(Forsyth et al., 2016). As was further discussed by Kilpua et al. (2020), the direct substorm injection to the ionosphere results in less intense precipitation than that driven by wave-particle interaction, which is in effect when the injected electrons are drifting into the morning sector after the injection. It was also concluded by Grandin et al. (2017) that strong CNA is more likely to occur during a substorm event with a longer duration.

Due to the much longer lifetime of recovery phases, as compared to the expansion phases, we found many more spacecraft overpasses during recovery phases than we did during the expansion phases. However, some basic conclusions can still be drawn from the available particle precipitation spectra. In Figure 7 we compare the particle precipitation spectra obtained in this study to the flux range constructed for pulsating aurora (PsA) by Tesema et al. (2020). Pulsating aurora, which occurs mainly towards the end of substorm recovery phases and beyond (Partamies et al., 2017), has been associated with high energy

precipitation. This includes significant electron fluxes at energies up to 200 keV (Miyoshi et al., 2015), which cause strong ionisation down to the lower D region ($\sim$70 km), and can consequently deplete the mesospheric ozone. In our results, the recovery phase fluxes are essentially PsA fluxes, although they are concentrated in the higher part of the PsA flux band. It was concluded by Tesema et al. (2020) that the low flux scenario of the PsA spectrum (bottom dashed curve in Figure 7) did not produce a chemical effect in the neutral D region atmosphere. It thus remains to be determined how to separate PsA

events that cause an atmospheric response from those for which the precipitation is too soft, and whether the low flux boundary of the recovery phase precipitation would correspond to the threshold for the atmospheric effects to happen. Furthermore, it is important to note that most of the recovery phase overpasses analysed here take place early in the recovery phases. We therefore speculate that the contribution of PsA to the recovery phase spectra shown in this study is small. Out of the 25 events covered by both optical and spacecraft data, 5 events showed optical signatures of PsA during the spacecraft overpass. If this is

a representative fraction for the whole dataset, it should not bias the median spectrum of all recovery phase overpasses. Overall the expansion phase spectra can be better described by the PsA upper envelope spectrum than its median up to energies of about 10 keV, while the recovery phase spectra tend to grow higher fluxes at energies above about 10 keV. A larger spacecraft dataset needs to be collected in the future to confirm these conclusions.



## 5    Conclusions

About 200 nights of substorm activity have been classified into categories of isolated, compound, and multi-night substorm events, while requiring a 2-day period of quiet time prior to each event. By comparing intensities of ground-magnetic deflections (local electrojet index) and cosmic radio noise absorption (CNA) measured in the same region, we conclude that the isolated substorm events rarely produce strong ionisation in the D region ionosphere (strong CNA). In addition, we have illustrated that the isolated substorm events have short durations and limited spatial extent as compared to compound and multi-night substorm events.

For multi-night substorm events preceded by magnetically quiet conditions, the CNA intensity typically grows during the first 1–3 nights of the sequence before reaching the maximum D region impact. Similar conclusions can be drawn from the space-borne measurements of precipitating particles, where the fluxes at high energies ($\sim$100 keV) increase significantly from the first night to the second night of substorm activity. The number of substorm intensifications (particle injections) also increases the high-energy electrons in the precipitating particle population.

As demonstrated by earlier studies, periods of continuous activity with recurring substorms can lead to significant ionisation in the lower ionosphere, which in turn causes significant mesospheric ozone depletion. These model results of the mesospheric ozone depletion were associated with a multi-day sequence of tens of substorms, which is far from the average long-term substorm rate of a few per day. We used the CNA observations to estimate the atmospheric effect of the different substorm types, and are inclined to conclude that the isolated events may not be important in a long-term perspective. This would explain why the geomagnetic indices have been good energetic particle precipitation proxies despite their poor temporal resolution and spatial coverage. A more systematic precipitation spectrum study is required to include/exclude events with certain intensities and lifetimes, but in order to get there, the spatial extent of high-energy precipitation during different substorm types needs to be studied in more detail.

*Data availability.* Global geomagnetic activity index data were obtained through Kyoto World Data Center (http://wdc.kugi.kyoto-u.ac.jp), and solar wind data has been downloaded from the OMNIWeb database (https://omniweb.gsfc.nasa.gov). DMSP data has been downloaded from the CEDAR madrigal database at (http://cedar.openmadrigal.org). MIRACLE ASC quicklook data are available at (https://space.fmi.fi/MIRACLE/ASC/), full resolution image data can be requested from FMI (kirsti.kauristie@fmi.fi).

*Author contributions.* The event selection and classification was done by EK for his MSc thesis (https://ntnuopen.ntnu.no/ntnu-xmlui/handle/11250/2656642). FT provided the pre-processed POES and DMSP data, and helped with the analysis. All authors have contributed to the discussion of the results and the writing of the paper.

*Competing interests.* The authors declare no conflict of interest.





*Acknowledgements.* The work by NP, FT & HNT is supported by the Norwegian Research Council (NRC) under CoE contract 223252, EH & EB by NRC under contract number 287427. NP, FT & HNT further acknowledge the Young CAS (Centre for Advanced Studies)

440  fellow program. We thank the institutes who maintain the IMAGE Magnetometer Array: Tromsø Geophysical Observatory of UiT the Arctic University of Norway (Norway), Finnish Meteorological Institute (Finland), Institute of Geophysics Polish Academy of Sciences (Poland), GFZ German Research Centre for Geosciences (Germany), Geological Survey of Sweden (Sweden), Swedish Institute of Space Physics (Sweden), Sodankylä Geophysical Observatory of the University of Oulu (Finland), and Polar Geophysical Institute (Russia). Sodankylä Geophysical Observatory is acknowledged for the riometer data. We thank K Kauristie, S. Mäkinen, J. Mattanen, A. Ketola, and C.-F. Enell

445  for maintaining MIRACLE camera network and data flow. We thank NOAA's SWPC and NCEI (formerly NGDC) for the availability of NOAA POES data.




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
