# Peer review of "Electron precipitation characteristics during isolated, compound and multi-night substorm events"

_Annales Geophysicae, 2020_

## Referee Comment (RC1) · Anonymous Referee #1 · 4 Oct 2020

In this paper, the authors intended to combine a list of auroral substorms, computed from local AL index values in Lapland, with observations of CNA and electron precipitation. The authors investigated how the characteristics of electron precipitation changes during episodes of substorms. The authors classified the events into three categories, isolated, compound and multi-night substorms and found that the isolated substorms are insufficient for causing low-altitude ionization. For the multi-night substorm cases, the CNA intensity had a maximum values at the later stage in the consecutive days which indicates that the impact on the D region ionization (and possible destruction of ozone) is not expected soon after the first onset. The statistical results obtained in this study are important for evaluating/predicting the impact of electron precipitation on

the middle atmosphere. The study is well organized and this reviewer recommends a prompt publication after considering some minor points listed below:

**Comments:**

1. The Introduction section nicely reviews the recent studies related to the evolution of magnetospheric/precipitating electrons during episodes of auroral substorm. However, it is a bit difficult to pick up "what is unknown in this research area? and what will be revealed in the current paper." This reviewer suggests the authors to pin-point the target of current study somewhere in the Introduction section.

2. I am confused of the difference between the IL and $IL_{asc}$ indices. My understanding is that IL is the local AL index made from the entire IMAGE network while $IL_{asc}$ is a similar local AL value but only with data from Lapland stations of IMAGE, is this correct?

3. In the current method, the expansion phase onset is defined as the start time of negative bay in the AL index. This reviewer well understands that this is the only possible way to identify the onset time from the AL time-series. At the same time, however, I suspect that this onset timing is slightly earlier than that of actual "optical" onset. Such a systematic delay can be seen in the examples in Figures 1 and 2. Do the authors have any discussion on the difference between the optical and magnetic onset?

4. There is a difference in the response of CNA between the case example of an isolated substorm in Figure 1 and the superposed-epoch-analysis one in Figure 4 (left). The CNA absorption has a maximum value at the minimum of AL in the case example, but it is largest at the expansion phase onset in the superposed-epoch-analysis one. Could the authors provide some comments on this difference somewhere in the manuscript?

5. Figure 7: How close was the overpass of the DMSP/NOAA satellite? I presume that the satellite obtained multiple spectra during one specific overpass. Did the authors simply integrate all the spectra and generated one representative one? I would just like to know how the spectra from the satellites were corrected and integrated.

6. What is the "boundary fluxes of the pulsating aurora"? This reviewer is just unable to understand the meaning of "boundary."

7. Is there any orbital bias in the MLT coverage of DMSP/NOAA overpasses? Some previous studies implied that the energy of precipitating electrons causing pulsating aurorae tends to be harder in the later MLT (i.e., in the late morning sector). In this study, if the satellites only cover local time sectors, say before 03 MLT, the flux during the recovery phase might have been underestimated.

---

## Referee Comment (RC2) · Anonymous Referee #2 · 22 Oct 2020

The authors use a combination of ground based and space based observations to investigate energy spectra and flux variations of precipitating electrons during expansion and recovery periods of substorms. The results are very interesting, but I recommend some clarifications to the paper before it is accepted for publication.

The DMPS and POES electron measurements are only presented one figure (Figure 7) and the discussion in section 3.2 focuses on contrasting the DMPS energy ranges to the pulsating aurora fluxes. What about the POES observations? Why does there appear to be a large discrepancy between the DMPS and POES fluxes between 1-10 keV energies? What about higher energies? Note that the value of the top energy

range is cut off from the left panel in Figure 7.

The authors relate the substorms under investigation to the potential atmospheric impacts throughout the paper. I note that the atmospheric impact in the paper is used to discuss both the ionisation impact and referring to $NO_x$ and $HO_x$ production/ozone loss and often it is initially not clear to the reader which one is meant. I recommend reading the text through carefully and clarifying where needed. Substorms are indeed one of the main unknowns in the existing proxies for electron precipitation particularly when considering the eV to tens of keV vs. hundreds of keV energy precipitation. Global atmospheric models that extend to the thermosphere usually include the lower energy range via parameterisation that is driven for example by the Kp-index, while the higher energies are implemented via a POES/MEPED based precipitation model that is driven by the Ap-index (for both, see Matthes, K. et al. (2017), Solar forcing for CMIP6 (v3.2), Geoscientific Model Development, 10(6), 2247–2302, doi:10.5194/gmd-10-2247-2017). The authors touch on this in the discussion section (lines 360-362) where they write in the context of their results that: *These findings emphasise the atmospheric influence of the compound and multi-night substorm events, which may explain why the global geomagnetic activity indices serve us so well as energetic particle precipitation proxies, despite their poor temporal and spatial resolution.* and again the in conclusions: *This would explain why the geomagnetic indices have been good energetic particle precipitation proxies despite their poor temporal resolution and spatial coverage.* Unfortunately I found no explanation or background of the use of global indices in proxies – for anyone unfamiliar with the details of electron precipitation proxies, this important result will likely be lost and thus I recommend revising the text to make the context clear. Can you comment on the relation of indices like Ap and Kp to the ones used in this study - this again would be useful context for the users of those global indices.

On several occasions the impacts on $NO_x$ and $HO_x$ production are referred to as this a direct atmospheric consequence of electron precipitation. For example in the discussion: *Compound and multi-night substorms will, however, be significant contributors to the direct production and $HO_X$ and $NO_X$ radicals in the atmosphere.* I agree that this might be the case, but a statement like this should either be backed up by suitable references or toned-down (e.g. *... likely be significant...*) as the present work does no involve any direct analysis of the production. Please check these aspects in the text and clarify where needed.

Lines 151-152: Are the locations of the Abisko and Sodankylä riometers the same as the magnetometers in section 2.1? No location is given for the two, only Ivalo.

Line 294: Wording check: Does this mean events with $IL < -300$ nT are excluded? or ones with $IL > -300$ nT?

Line 306: First mention of HSS?

Lines 370-373: In the context of the atmospheric impact, do you mean when contrasted to the likely impact from the expansion and recovery phases?

Line 380: What is M-I?

Line 396: *deplete the mesospheric ozone* indicated the process is destroying all mesospheric ozone. Suggest *cause depletion of mesospheric ozone* instead.

---

## Author Comment (AC1) · 13 Nov 2020

We thank the referee for careful reading of the manuscript and raising important items for further clarification. Below are our responses to each comment, with comments in *italic*.

Point-by-point responses:

1. *The Introduction section nicely reviews the recent studies related to the evolution of magnetospheric/precipitating electrons during episodes of auroral substorm.*

[Figure]

*However, it is a bit difficult to pick up "what is unknown in this research area? and what will be revealed in the current paper." This reviewer suggests the authors to pin-point the target of current study somewhere in the Introduction section.*

The aim of the current study needs to be clear. It is the question of the precipitation during different types of substorms and during the evolution of a substorm, which has not been addressed by previous studies. At the end of the introduction this will be phrased as: "Our aim with this study is to investigate whether all substorms are equally influential to the neutral atmosphere, or whether the intensity, duration or internal structure of the substorms can be used to differentiate events which have a significant atmospheric impact from those which only have a negligible impact."

2. *I am confused of the difference between the IL and $IL_{asc}$ indices. My understanding is that IL is the local AL index made from the entire IMAGE network while $IL_{asc}$ is a similar local AL value but only with data from Lapland stations of IMAGE, is this correct?*

This is correct. IL is the local AL index based on data from the entire IMAGE network, and $IL_{asc}$ only includes data from the 5 Lapland auroral camera stations. This is explained at the end of the section 2.1. For brevity we use the term "IL index" to refer to $IL_{asc}$, which may cause some confusion but we will re-phrase this in a clearer way in the revised version.

3. *In the current method, the expansion phase onset is defined as the start time of negative bay in the AL index. This reviewer well understands that this is the only possible way to identify the onset time from the AL time-series. At the same time, however, I suspect that this onset timing is slightly earlier than that of actual "optical" onset. Such a systematic delay can be seen in the examples in Figures 1 and 2. Do the authors have any discussion on the difference between the optical and magnetic onset?*

Absolutely correct. There is a small temporal difference between the "magnetic

onset" and the "optical onset" or the auroral breakup. This was discussed by Partamies et al. (2015) where they concluded that the time delay is typically of the order of a minute. As their result was based on optical aurora and the changes in the auroral structures (from arcs to more complex forms), the time delay may not be exactly the same between the magnetic onset and the onset of energetic precipitation (seen as CNA). This will be mentioned in the new version of the manuscript in connection to Figure 2.

4. *There is a difference in the response of CNA between the case example of an isolated substorm in Figure 1 and the superposed-epoch analysis one in Figure 4 (left). The CNA absorption has a maximum value at the minimum of AL in the case example, but it is largest at the expansion phase onset in the superposed-epoch analysis one. Could the authors provide some comments on this difference somewhere in the manuscript?*

This is a good point. The example events in Figure 1 and 2 are plotted with 1-min temporal resolution, while the superposed epoch curves have an hourly resolution. Since the expansion phases are often short (less than an hour), the epoch evolution shows the maximum CNA at the epoch onset. This temporal resolution issue will be clarified in the revised text: "Note that the hourly resolution of the superposed epoch analysis places the maximum CNA values at the onset hour, although in higher resolution data they tend to occur around the minimum IL time, as seen in Figure 1."

5. *Figure 7: How close was the overpass of the DMSP/NOAA satellite? I presume that the satellite obtained multiple spectra during one specific overpass. Did the authors simply integrate all the spectra and generated one representative one? I would just like to know how the spectra from the satellites were corrected and integrated.*

The DMSP/NOAA overpasses were searched within the common FoV of the Lapland ASCs and all the spectra are overpass-averaged spectra. These things will

be clarified in the new version of the manuscript: "Each spectrum in the figure is an average over an individual overpass, where an overpass is defined as a conjugate with the common field-of-view of Lapland auroral cameras, as described by Tesema et al. (2020)."

6. *What is the "boundary fluxes of the pulsating aurora"? This reviewer is just unable to understand the meaning of "boundary."*
That is indeed a confusing statement. The boundary fluxes are the upper and lower envelope flux curves for the pulsating aurora as determined in the statistical study of Tesema et al. (2020). We will use this more precise terminology in the new version.

7. *Is there any orbital bias in the MLT coverage of DMSP/NOAA overpasses? Some previous studies implied that the energy of precipitating electrons causing pulsating aurorae tends to be harder in the later MLT (i.e., in the late morning sector). In this study, if the satellites only cover local time sectors, say before 03 MLT, the flux during the recovery phase might have been underestimated.*
There is an MLT bias of DMSP/NOAA overpasses, as described by Tesema et al. (2020), where they suggest that the hardening of the precipitation only takes place after about 06:30 MLT and is largely due to the decay of softer precipitation. It is unlikely to observe recovery phases that late in MLT. It is rather only pulsating aurora that is often seen after about 4 MLT, and during most of those events the magnetic deflection has already recovered. This comment will be added in the discussion.

---

## Author Comment (AC2) · 13 Nov 2020

We thank the referee for careful reading and thoughtful comments on the manuscript. Below are our responses to each comment, with comments in *italic*.

Point-by-point responses:

- *The DMSP and POES electron measurements are only presented one figure (Figure 7) and the discussion in section 3.2 focuses on contrasting the DMSP energy ranges to the pulsating aurora fluxes. What about the POES observa-*

[Figure]

*tions? Why does there appear to be a large discrepancy between the DMSP and POES fluxes between 1–10 keV energies? What about higher energies? Note that the value of the top energy range is cut off from the left panel in Figure 7.*

The figure discussion focusses on the comparison of the substorm energy fluxes with the pulsating aurora fluxes because that is our "known" reference at the moment. The discrepancy between the DMSP and POES fluxes at energies of 1–10 keV is most likely due to the fact that the two spacecraft are mainly not measuring the same events. We have simply just collected all data from overpasses of the two spacecraft during any of the analysed substorm expansion and recovery phases, so very few of them are time-wise close to each other. Thus, discontinuities are expected from one spacecraft measurement to another, both at low and high energies. This will be commented in the revised version for clarity. The top energy range of the left panel will be made visible.

- *The authors relate the substorms under investigation to the potential atmospheric impacts throughout the paper. I note that the atmospheric impact in the paper is used to discuss both the ionisation impact and referring to $NO_x$ and $HO_x$ production/ozone loss and often it is initially not clear to the reader which one is meant. I recommend reading the text through carefully and clarifying where needed.*

The atmospheric impact we are interested in is the production of $NO_x$ and $HO_x$, which then leads to depletion of mesospheric ozone. The ionisation impact, which is also talked about, is just what we can measure as an enhanced CNA. So, strong ionisation is considered equal to the production of the catalysts, which is why the "impact" may refer to either process. We will emphasise this logic in the revised version of the introduction: "In this paper, we use the cosmic noise absorption enhancement as a measure for the medium-energy electron precipitation, which have the potential to produce odd hydrogen and odd nitrogen and thus lead to depletion of mesospheric ozone."

- *Substorms are indeed one of the main unknowns in the existing proxies for electron precipitation particularly when considering the eV to tens of keV vs. hundreds of keV energy precipitation. Global atmospheric models that extend to the thermosphere usually include the lower energy range via parameterisation that is driven for example by the Kp-index, while the higher energies are implemented via a POES/MEPED based precipitation model that is driven by the Ap-index (for both, see Matthes, K. et al. (2017), Solar forcing for CMIP6 (v3.2), Geoscientific Model Development, 10(6), 2247–2302). The authors touch on this in the discussion section (lines 360–362) where they write in the context of their results that: "These findings emphasise the atmospheric influence of the compound and multi-night substorm events, which may explain why the global geomagnetic activity indices serve us so well as energetic particle precipitation proxies, despite their poor temporal and spatial resolution." and again the in conclusions: "This would explain why the geomagnetic indices have been good energetic particle precipitation proxies despite their poor temporal resolution and spatial coverage." Unfortunately I found no explanation or background of the use of global indices in proxies — for anyone unfamiliar with the details of electron precipitation proxies, this important result will likely be lost and thus I recommend revising the text to make the context clear. Can you comment on the relation of indices like Ap and Kp to the ones used in this study — this again would be useful context for the users of those global indices.*

Some more background on magnetic index proxies is certainly needed. The revised discussion text will say:

"The atmospheric and climate models currently use Kp and Ap index based proxies to describe the ionisation rates due to energetic electron precipitation

(Matthes et al., 2017). As the temporal resolution of these global indices is three hours, a short-lived (less than 3 hours) isolated substorm may only result in a minor magnetic variation with respect to its maximum magnetic deflection in an electrojet index data (used in this study), which is available in 1-min resolution. Thus, the current energetic electron precipitation proxies are likely to only describe well the longer-lasting compound and multi-night type substorm events. Our findings emphasise the atmospheric influence of the compound and multi-night substorm events, which may explain why the global geomagnetic activity indices serve us so well as energetic particle precipitation proxies, despite their poor temporal and spatial resolution"

"This would explain why the geomagnetic Kp and Ap indices have been good energetic particle precipitation proxies in climate models despite their poor temporal resolution and spatial coverage."

- *On several occasions the impacts on $NO_x$ and $HO_x$ production are referred to as a direct atmospheric consequence of electron precipitation. For example in the discussion: "Compound and multi-night substorms will, however, be significant contributors to the direct production and $HO_x$ and $NO_x$ radicals in the atmosphere." I agree that this might be the case, but a statement like this should either be backed up by suitable references or toned-down (e.g. . . . "likely be significant". . . ) as the present work does not involve any direct analysis of the production. Please check these aspects in the text and clarify where needed.*

Since no chemical modelling or observations are done in this study this is a good point. This particular statement in the discussion will be backed up by referring to Seppälä et al. (2015) who presented ion chemistry model results for a series of substorms, which corresponds to our multi-night substorm events. Those results showed significant mesospheric ozone loss due to production of $HO_x$ and $NO_x$ radicals.

- *Lines 151–152: Are the locations of the Abisko and Sodankylä riometers the same as the magnetometers in section 2.1? No location is given for the two, only Ivalo.*
  The locations of the Abisko and Sodankylä riometers are indeed the same as the magnetometers, which is why no new locations were given. But the fact that they are co-located will be mentioned in the revised version for clarity.

- *Line 294: Wording check: Does this mean events with IL<-300 nT are excluded? or ones with IL>-300 nT?*
  This means that events with IL>-300 nT are excluded. The new version will say: "deflections smaller than 300 nT", which should leave less room for misunderstanding.

- *Line 306: First mention of HSS?*
  Yes, the revised version will read: "high-speed stream"

- *Lines 370–373: In the context of the atmospheric impact, do you mean when contrasted to the likely impact from the expansion and recovery phases?*
  This is indeed a statement comparing to the likely impact from the expansion and recovery phases, and will be re-written to: "They are, therefore, unlikely to sum up to a large global atmospheric impact in contrast to the expansion and recovery phases."

- *Line 380: What is M–I?*
  The revised version will read as: "magnetosphere–ionosphere"

- *Line 396: "deplete the mesospheric ozone" indicated the process is destroying all mesospheric ozone. Suggest "cause depletion of mesospheric ozone" instead.*
  The phrase will be changed accordingly.